# Effects of Mixed Fermentation on the Aroma Compounds of 'Italian Riesling' Dry White Wine in Eastern Foothill of Helan Mountain

Hongchuan Xia [1,2,†] , Zhong Zhang [3,†] , Lijun Sun [1], Qingchen Zhang [4] and Junxiang Zhang [5,6,*]

1 School of Agriculture, Ningxia University, Yinchuan 750021, China; hongchuanxia@126.com (H.X.)
2 Yinchuan Wine Industry Development Service Center, Yinchuan 750021, China
3 School of Life Sciences, Ningxia University, Yinchuan 750021, China
4 College of Pharmacy, University of Florida, Gainesville, FL 32610, USA
5 School of Food and Wine, Ningxia University, Yinchuan 750021, China
6 China Wine Industry Technology Institute, Yinchuan 750021, China
* Correspondence: zhangjunxiang@126.com; Tel.: +86-138-9501-3338
† These authors contributed equally to this work.

**Abstract:** To study the effect of mixed fermentation of non-*Saccharomyces* strains and *Saccharomyces cerevisiae* on the aroma quality of 'Italian Riesling' wine in the eastern foothill of Helan Mountain and to determine the most optimum process of mixed fermentation, two selected non-*Saccharomyces* strains, including *Hanseniaspora uvarum* YUN268 and *Pichia fermentans* Z9Y-3, were inoculated with *Saccharomyces cerevisiae* in different proportions (10:1 or 1:1) and different stages (48 h in advance or simultaneously at the beginning) to ferment 'Italian Riesling' dry white wine. The oenological parameters and aroma indexes of the wine samples were evaluated. The results showed mixed fermentation can not only reduce the alcohol content of wine 0.24~0.71% vol but also increase the glycerol content to improve the taste of wine. The mixed fermentation effect of *Pichia fermentans* Z9Y-3 and *Saccharomyces cerevisiae* resulted in improvements, especially the high proportion (10:1) sequential inoculation and simultaneous inoculation of wine samples (WSP10 and WCP10), which not only produced more volatile aroma substances and glycerol content but also increased the total amount of ester substances by 49.4% and 56.5%, respectively, compared with the control. The sensory evaluation scores of WSP10 and WCP10 were significantly higher than the control (89.3 and 88.1 points, respectively). At the same time, it can also enhance the aroma of lemon, cream, almond, and others and increase the aroma complexity of wine. Therefore, these two methods of mixed fermentation inoculation are more suitable for the production of Italian Riesling wine in the eastern foothill of Helan Mountain. In conclusion, the mixed fermentation of *Pichia fermentans* Z9Y-3 and *Saccharomyces cerevisiae* 10:1 (simultaneous or sequential) inoculation is suitable for the production of Italian Riesling dry white wine in the eastern foothill of Helan Mountain.

**Keywords:** *Hanseniaspora uvarum*; *Pichia fermentans*; aroma compounds; sensory quality; mixed fermentation; sequential fermentation

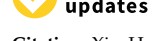



## 1. Introduction

*Saccharomyces cerevisiae* is characterized by excellent oenological performances and is widely employed in the industrial production of grape wine [1]. However, the rapid dominance of this yeast species during alcohol fermentation (AF) often hinders the activities of other microorganisms originating from vineyards and wineries, thereby leading to homogeneity in wine product styles [2]. Given the growing demand for product diversity in the wine market, people are exploring novel solutions to tackle this challenge. Research has indicated that numerous non-*Saccharomyces* (NS) yeasts produce higher amounts of volatile secondary metabolites than *S. cerevisiae* and can secrete glycosidases to facilitate the release

of bonded aromatic compounds in grapes, thus playing a significant role in the formation of wine fragrance [3]. Despite their potential benefits, some NS yeasts have historically been considered undesirable microorganisms for elevating the concentrations of acetic acid, volatile phenols, and hydrogen sulfide, which are detrimental to wine quality [3,4]. Furthermore, NS yeasts often struggle to adapt to the fermentative environment and cause fermentation termination or stagnation [5].

The mixed-culture fermentation technique, also known as co-fermentation, offers a promising solution to the issues mentioned above. It involves intentionally introducing two or more wine yeasts into grape juice or must in a specific ratio, either simultaneously or sequentially, to fully capitalize on their advantages and enable them to collaborate to finish the AF process [6]. The use of selected multi-strain starters for controlled mixed fermentation was first proposed by Amerine as early as the 1960s [7]. Castelli et al. (1955) recommended the use of a combination of *S. cerevisiae* and *Torulaspora delbrueckii* to decrease the acetic acid content [8]. In 1979, Snow et al. suggested sequential inoculation of *Schizosaccharomyces pombe* and *S. cerevisiae* help to consume malic acid in wine [9]. Nevertheless, the evidence presented by Mora et al. (1990) and Kapsopoulou et al. (2007) demonstrated that co-fermentation of *Kluyveromyces thermotolerans* and *S. cerevisiae* exhibited an acidification effect with an increase in wine acidity of up to 70% [10,11]. Additionally, mixed fermentation consisting of NS yeast can reduce the ethanol content and enhance the glycerol and color stability of wine [12]. Recent research in mixed fermentation has prioritized modifying the wine flavor. *Hanseniaspora uvarum*, *Pichia fermentans*, *Metschnikowia pulcherrima*, *Pichia kluyveri*, and *T. delbrueckii* are some of the NS yeasts commonly utilized for this purpose [13–16]. These yeasts have the ability to amplify the production of both primary aromas (terpenes, C13-norisoprenoids, and thiols) [13,14] and secondary aromas (esters, aromatic alcohols, and fatty acids) in wine [12,17–19].

The eastern foot of Helan Mountain in Ningxia has emerged as a prominent wine region in the global wine atlas where *Vitis vinifera* L. cv. Italian Riesling is one of the main white cultivars planted [20]. Despite having an intense flavor and a soft taste, the aroma of Italian Riesling wine produced in Ningxia is of limited typicality and indistinguishable from that of wines made in other regions. To overcome this problem, our group introduced two preferred NS yeast strains (*H. uvarum* and *P. fermentans*) and conducted mixed fermentations using various inoculation schemes alongside purely inoculated *S. cerevisiae* fermentation. Through our investigations, we examined the alterations in basic oenological indicators, volatile organic compounds (VOCs), and olfactory profiles of wine. Our results led us to propose a viable mixed inoculation plan for local 'Italian Riesling' wine production.

## 2. Materials and Methods

### 2.1. Grapes

Italian Riesling grapes were collected from Ganchengzi vineyard (Qingtongxia, Ningxia Hui Autonomous Region, China) on September 7, 2020. The vines are 5 to 8 years old, reduced sugar is 227.5 g/L, specific gravity is 1.099, titrating acid is 5.4 g/L (as tartaric acid), and pH is 3.56 in a good sanitary condition.

### 2.2. Strains and Media

*S. cerevisiae* (Excellence TXL) was purchased from LAMOTHE ABIET; *H. uvarum* [21] (Yun268), and *P. fermentans* [22] (Z9Y-3) were all provided by the Wine School of Northwest Agriculture and Forestry University. The Wallerstein Laboratory nutrient agar medium, also known as WLN medium, was initially used to monitor yeast populations during fermentation. In this experiment, the WLN medium can be used to quickly identify most of the relevant yeast by the characteristic morphology of each yeast colony and classify each yeast. Shown in Table 1 are the morphological characteristics of the yeast used on the WLN medium in this study [23].

**Table 1.** Colony morphology of yeasts.

| Strain Number | Species | Colony Color [24] | Colony Features [24,25] | Colony Map |
|---|---|---|---|---|
| Excellence TXL | *Saccharomyces cerevisiae* | Cream, ribbon green | Specular protrusion, smooth, regular surface, and opaque |  |
| Yun268 | *Hanseniaspora uvarum* | Bottle green | Flat, smooth, opaque, and oily |  |
| Z9Y-3 | *Pichia fermentans* | Cream | Indicating wrinkled and opaque, serrated edges, round |  |

Yeast extract peptone dextrose (YEPD) medium and WLNmedium were purchased from High-tech Industrial Park Haibo Biotechnology Co., Ltd. (Qingdao, China). Pectinase (Vinozym vintage FCE), potassium metabisulfite, bentonite, and polyvinylpyrrolidone (PVPP) were purchased from LAMOTHE ABIET (Yinchuan, China). Anhydrous ethanol (chromatographic pure) was purchased from Sinopharm Group Chemical Reagent Co., Ltd. (Shanghai, China), C8~C20 normal alkane (99.7%, GC) was purchased from Sigma-Aldrich Corporation (Shanghai, China), and 4-Methyl-2-pentanol (98.0%, GC) was purchased from TCI Corporation (Shanghai, China).

*2.3. Fermentation Strategies*

The traditional process is used to make dry white wine, which is divided into seven fermentation steps, as shown in Table 2. The yeast inoculation program is shown in Table 3.

**Table 2.** Fermentation strategies.

| Fermentation Step | Fermentation Strategies |
|---|---|
| Step1 | The grape was crushed and pressed to extract the juice in a ton-controlled temperature fermenter (50 mg/L $SO_2$, 20 mg/L pectinase was added). |
| Step2 | Adding 1.0 g/L bentonite and 100 mg/L PVPP for clarification treatment. |
| Step3 | Impregnating at low temperature (4 °C) for 24 h, then separating the clarification juice in a 5 L glass fermenter. Finally, return temperature to 18 °C. |
| Step4 | The yeast was inoculated according to different groups. A total of eight experimental treatments were set up (Table 2). |
| Step5 | The temperature was controlled at 18–22 °C for alcohol fermentation. Determined the specific gravity of grape juice every two days in the fermentation process. |
| Step6 | Added 60 mg/L $SO_2$ at the end of fermentation and transferred the wine into a clean and hygienic 5 L glass jar for sealed storage. |
| Step7 | Carried out normal clarification and stability before bottling for storage. The wine was analyzed a month later. |

**Table 3.** Yeast inoculation program.

| Control | Mixed Fermentation | | | | | | | |
| --- | --- | --- | --- | --- | --- | --- | --- | --- |
| *S.cerevisiae* | *H. uvarum & S.cerevisiae* | | | | *P. fermentans & S.cerevisiae* | | | |
| CK<br>*S. cerevisiae* was inoculated alone | WCH1<br>simultaneous Inoculation, (1:1) | WCH10<br>simultaneous Inoculation, (10:1) | WSH1<br>Sequential inoculation, (1:1) | WSH10<br>Sequential inoculation, (10:1) | WCP1<br>simultaneous inoculation, (1:1) | WCP10<br>simultaneous Inoculation, (10:1) | WSP1<br>Sequential inoculation, (1:1) | WSP10<br>Sequential inoculation (10:1) |

Note: CK is only inoculated commercial *S. cerevisiae* $1 \times 10^6$ CFU/mL. NS inoculation 1:1 ($1 \times 10^6$ CFU/mL), 10:1 ($1 \times 10^7$ CFU/mL), simultaneous inoculation means adding *S. cerevisiae* and NS at the same time. Sequential inoculation means adding NS first and *S. cerevisiae* 48 h later.

### 2.4. Activation Culture of Strain

Non-*Saccharomyces* strains were cultivated in YPD solid medium at 28 °C for 24 h. Strains with distinct colony characteristics were purified using a repetitive streaking method. Then, the purified yeast strains were activated with YPD liquid medium for 24 h, mixed with 40% sterile glycerol at 1:1, and stored at −20 °C. Before use, the D1/D2 region sequence of 26S rRNA was identified again to confirm the strain for reserve [21,26].

The reserved strains were inoculated into 200 mL of YPD liquid medium and incubated at 28 °C and 180 r/min until the logarithmic growth phase. The cells were collected by centrifugation at 4 °C and washed three times with sterile water before being counted using a hemocytometer. The cells were then inoculated into grape juice in different protocols [26].

### 2.5. Yeast Count

During fermentation, wine samples were diluted in decimal series and streaked on WLN solid medium every two days. The growth of yeast was monitored by observing the distinct colony morphologies of *S. cerevisiae* and non-*Saccharomyces* on the WL plates [23].

### 2.6. Basic Oenological Parameters Analysis

Basic oenological parameter indexes were determined according to GB/T 15038-2006, China [27]. The alcohol content was measured using the densimeter method. Briefly, non-volatile substances were removed from wine through distillation, and the density of the distilled liquid was measured using a densitometer, which was then checked in a reference table in GB/T 15038-2006. Volatile acids and titratable acids were determined using the acid–base titration principle with phenolphthalein as an indicator, and the acid content was calculated based on the amount of sodium hydroxide used. pH was measured by magnetic pHS-3C pH meter; residual sugar and glycerol content were determined by Enology Y15 fully automatic wine analyzer (BioSystems, Barcelona, Spain).

### 2.7. Volatile Analysis

A 5 mL aliquot of undiluted wine was transferred into a 20 mL headspace vial containing 1.5 g of sodium chloride. To serve as an internal standard, 10 μL of 4-methyl-2-pentanol (1.0083 g/L) was added, resulting in an in-vial concentration of 2.01 mg/L. The vial was sealed using a magnetic PTFE/Sil cap and incubated at 40 °C for 5 min. Volatile compounds were extracted using a DVB/CAR/PDMS fiber (50/30 μm, 1 cm) for 30 min at 40 °C with continuous stirring at 250 r/min and desorbed in splitless mode at 240 °C for 10 min. The GC oven temperature was initially set at 40 °C for 3 min, followed by an increase to 97 °C and stayed for 7 min, a ramp of 2 °C/min up to 120 °C, a further increase of 3 °C/min to 150 °C, and finally, an increase of 8 °C/min up to 220 °C, which was held for 10 min. The transfer line temperature was maintained at 230 °C. The MS electron impact mode was utilized with an electron ionization source temperature of 230 °C and an electron energy of 70 eV. The solvent delay was set to 4.4 min [28].

Through qualitative and quantitative analysis of volatile substances in the scan mode, the volatile component mass spectrometry was extracted and retrieved through the NIST 17 spectrum library and according to the retention time of C8–C20 alkanes mixture standard and



the retention index method (Retention Index, RI) [29]. The concentration of aroma composition was calculated by the ratio of the peak area and then by the internal standard concentration.

### 2.8. Sensory Analysis

The wine sensory evaluation team was composed of wine professional teachers and ten students from Ningxia University (four females and six males). Each taster received professional evaluation training. In the form of blind tasting, if the standard wine glass (ISO 3591-1997) is used in the standard tasting room (ISO 8589-1998), the taster should score the wine quality according to the hundred percent sensory evaluation system developed by the International Organisation of Vine and Wine and select appropriate words from the Davis aroma wheel to describe the wine aroma characteristics [30].

### 2.9. Statistical Analysis

Data statistics were performed using Office 2016 software (Microsoft Office, Redmond, WA, United States). One-way ANOVA and the Duncan test were applied to determine the variance of basic oenological parameters, volatile aroma components, and sensory scores. The line diagram was drawn with Origin 2017 software (OriginLab Corporation, Northampton, MA, United States); the cluster heatmap was drawn with the 'Pheatmap' package in R software (version 3.6.1, R Foundation for Statistical Computing, Vienna, Austria).

## 3. Results and Discussion

### 3.1. Analysis of Alcohol Fermentation Rate

The fermentation curves for different experimental treatments are shown in Figure 1, and it can be found that all wine samples completed the fermentation in 16 to 20 days. Different mixed fermentation methods have great impacts on the fermentation rate of wine, as shown in Figure 1(A1); WCH1 and WCH10 completed the fermentation two to four days earlier than the other groups. Among them, the fastest fermentation was WCH10, which dropped to 0.994 at 16 days. In contrast, the slowest fermentation was the 1:1 sequential inoculation (WSH1), which completed the alcoholic fermentation in twenty days, four days slower than WCH10. Compared with the control, the species *H. uvarum* inoculated with *S. cerevisiae* can promote the process of alcohol fermentation. Secondly, the faster the fermentation, the more was added. These phenomena indicate the interaction between *H. uvarum* and *S. cerevisiae* favors alcoholic fermentation. As shown in Figure 1(A2), the pattern of fermented *P. fermentans* was more consistent with that of *H. uvarum*. The wine sample of 10:1 simultaneous inoculation (WCP10) finished the alcohol fermentation first, and the wine sample of 1:1 sequential inoculation (WSP1) finished slowest.

Comprehensive analysis showed *S. cerevisiae* is the species that mainly led the fermentation. In contrast with CK, the total number of yeasts increased after the addition of NS, thus accelerating the fermentation rate in the early stage. During the sequential inoculation of fermentation and since only NS was present in the first two days, the fermentation rate was slowly accelerated until *S. cerevisiae* was added after 48 h of fermentation. As shown in Figure 1, a high proportion of sequentially inoculated fermentation regimens, WSH10 and WSP10, had the same fermentation extent as the control on days 14 and 8, respectively. Although the fermentation was delayed for two days in the early stage of sequential inoculation, the fermentation rate was increased after the inoculation of *S. cerevisiae*; thus, it can be speculated that the co-inoculation of NS and *S. cerevisiae* can accelerate the alcoholic fermentation rate.

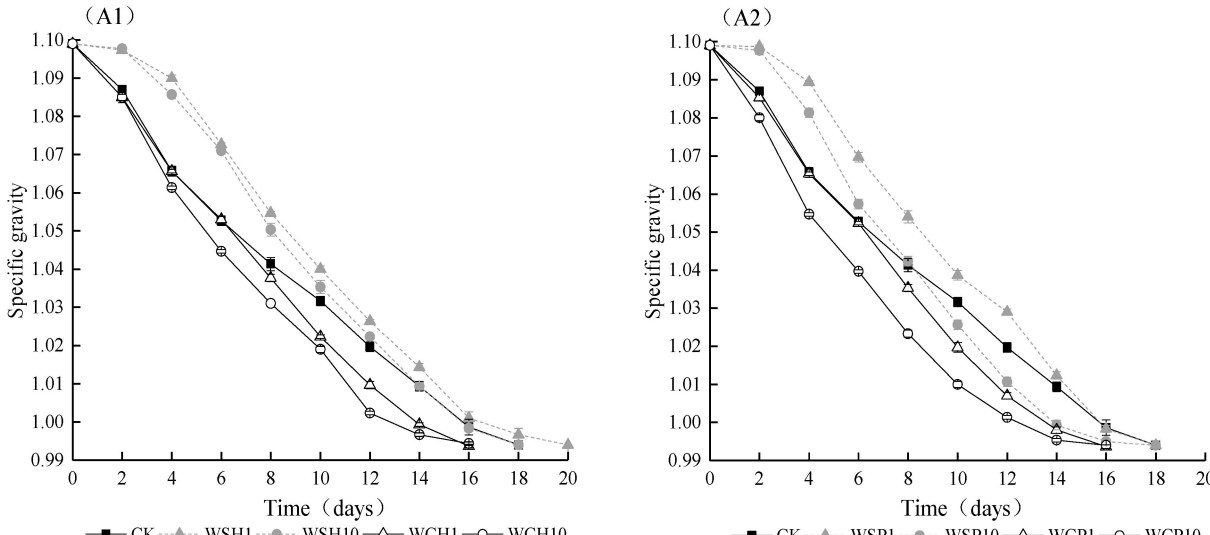

**Figure 1.** Fermentation curves of different treatments: (**A1**) inoculation of *Hanseniaspora uvarum*, (**A2**) inoculation of *Pichia fermentans*.

### 3.2. Growth Kinetics of Yeast Strains during Fermentations

During wine alcohol fermentation, wine samples were sampled and coated on WLN plates every four days. The dynamics of various yeast types during fermentation were monitored by the number of yeast colonies in WLN. The resulting yeast growth dynamics change curves are shown in Figure 2. As shown in (A1), *H. uvarum* survived in wine from 12 to 20 days due to different inoculation methods. The lower proportion of 1:1 simultaneously inoculated fermentation methods survived for the shortest time and could not be found on the WLN medium at 12 d. However, in the high proportion of the 10:1 sequential inoculation method, it can survive in the fermentation broth for a long time before gradually disappearing at 16 to 20 days. Overall, in the sequential inoculation system, the number of *H. uvarum* increased in the first two days and decreased after inoculation with *S. cerevisiae*, and the maximum number was up to $3.1 \times 10^6$ CFU/mL (WSH1) and $2.53 \times 10^7$ CFU/mL (WSH10). During simultaneous inoculation, the number of *H. uvarum* declined and disappeared earlier in the fermentation system earlier. As shown in Figure 2(B1), the effect of different inoculation methods on *P. fermentans* was similar to that of fermented yeast in *H. uvarum*, but the difference was that the survival time of *P. fermentans* was less than that of *H. uvarum*, with only 8–16 days.

According to Figure 2(A2), the *H. uvarum* had stronger activity in wine, which led to the inhibited growth of *S. cerevisiae* and eventually in the number of *S. cerevisiae* with different inoculation modes, but in Figure 2(B2), the number of *S. cerevisiae* in the fermentation system was not different, which can further show the competitiveness of fermented *S. cerevisiae* is less. The changing pattern of *S. cerevisiae* was more consistent, both rising first, reaching the maximum around the eighth day of fermentation, then decreasing, and finally, stabilizing at a higher order of magnitude ($10^6$–$10^7$) until the end of fermentation. *S. cerevisiae* in the controls grew best, reaching a maximum number at eight days of $4.17 \times 10^7$ CFU/mL.

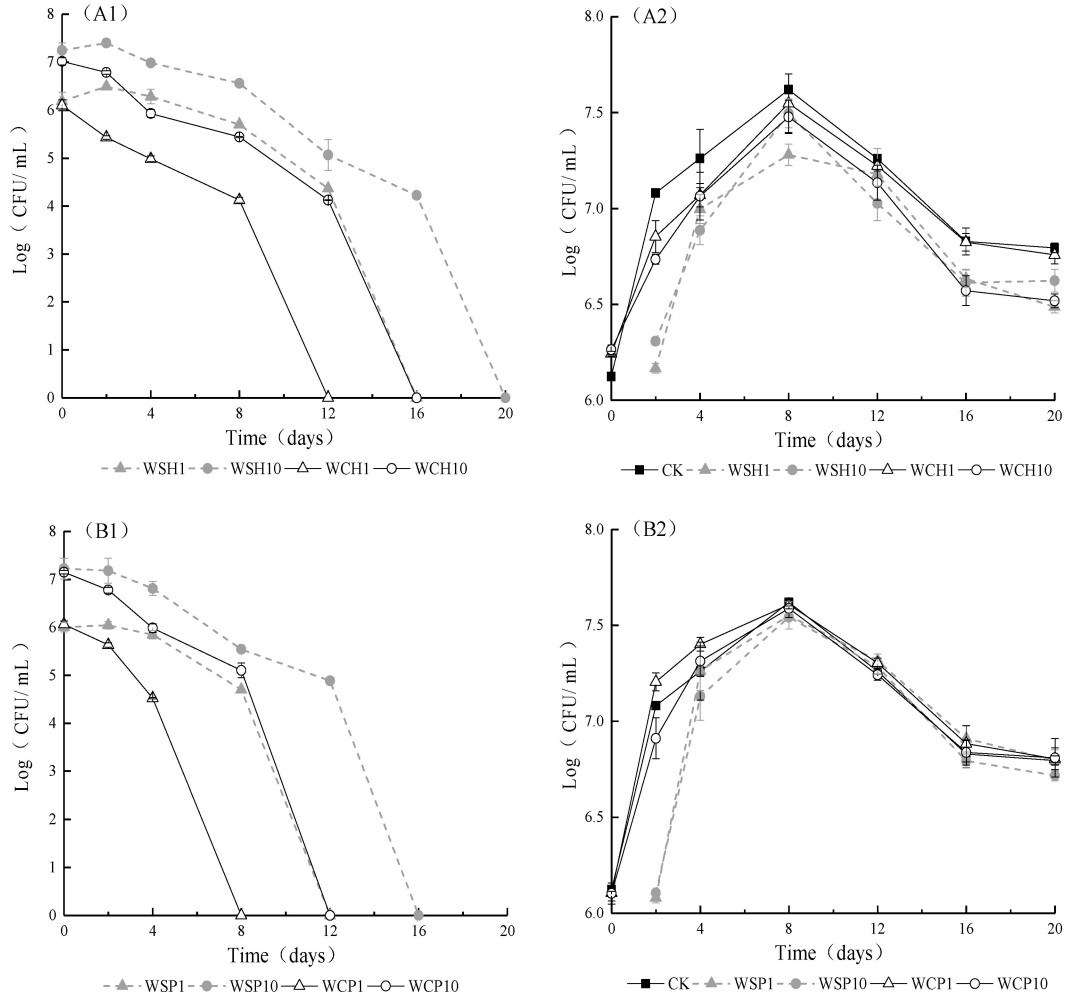

**Figure 2.** Growth status of *Saccharomyces cerevisiae* and NS yeasts in different fermentation systems: (**A**) inoculation of *Hanseniaspora uvarum*, (**B**) inoculation of *Pichia fermentans.* The 1 stands for NS, and 2 stands for *Saccharomyces cerevisiae*.

### 3.3. Wine Characteristics

The basic parameters of wine samples for each mixed fermentation and pure *Saccharomyces cerevisiae* culture are shown in Table 4. The residual sugar content of each inoculation treatment measured after inoculation was less than 4 g/L, which was consistent with the monitoring of the kinetics of wine fermentation (the specific gravity of all wines was finally below 0.994), which meant alcoholic fermentation could be completed in all fermentation schemes. After fermentation, measured alcohol content ranged from 11.61 to 12.32% vol. Residual sugar content ranged from 2.38 to 3.38 g/L. Titratable acidity content (measured by tartaric acid) ranged from 4.8 to 5.36 g/L. pH ranged from 3.51 to 3.65. Glycerol content ranged from 7.35 to 7.82 g/L. Volatile acid content ranged from 0.28 to 0.66 g/L.

**Table 4.** Basic oenological parameters of wine samples.

| Parameters | Control | *Hanseniaspora uvarum* | | | | *Pichia fermentans* | | | |
| --- | --- | --- | --- | --- | --- | --- | --- | --- | --- |
| | CK | WSH1 | WSH10 | WCH1 | WCH10 | WSP1 | WSP10 | WCP1 | WCP10 |
| Ethanol (% vol) | 12.32 ± 0.08 a | 11.95 ± 0.03 b | 11.65 ± 0.03 de | 12.06 ± 0.01 b | 11.74 ± 0.09 cd | 11.8 ± 0.06 c | 11.61 ± 0.02 e | 11.96 ± 0.02 b | 11.79 ± 0.02 c |
| Residual sugar (g/L) | 2.88 ± 0.18 b | 3.25 ± 0 ab | 3.13 ± 0.18 ab | 3.13 ± 0.18 ab | 2.38 ± 0.18 c | 3 ± 0 ab | 2.88 ± 0.18 b | 3.38 ± 0.18 a | 2.88 ± 0.18 b |
| Titratable acidity (g/L) | 4.99 ± 0.26 ab | 5.17 ± 0 ab | 5.36 ± 0.26 a | 4.8 ± 0 b | 5.17 ± 0 ab | 5.17 ± 0 ab | 4.99 ± 0.26 ab | 5.17 ± 0 ab | 5.36 ± 0.26 a |
| pH | 3.65 ± 0.01 b | 3.54 ± 0.01 e | 3.51 ± 0.01 f | 3.69 ± 0.01 a | 3.54 ± 0.01 e | 3.54 ± 0.01 e | 3.63 ± 0.01 c | 3.56 ± 0.01 d | 3.52 ± 0.01 f |
| Glycerol (g/L) | 7.35 ± 0.01 f | 7.63 ± 0.01 de | 7.68 ± 0.02 c | 7.6 ± 0.02 e | 7.73 ± 0 b | 7.78 ± 0.03 a | 7.82 ± 0.01 a | 7.66 ± 0.01 cd | 7.73 ± 0.02 b |
| Volatile acidity (g/L) | 0.28 ± 0.02 f | 0.51 ± 0.02 b | 0.66 ± 0.02 a | 0.29 ± 0.03 f | 0.43 ± 0.01 c | 0.33 ± 0 e | 0.37 ± 0.01 de | 0.33 ± 0.01 e | 0.38 ± 0.01 d |

Note: The data are presented in the form of 'mean ± standard deviation'. In the rows, different letters represent significant differences between treatments (Duncan's test, $p < 0.05$).

Compared with the CK, the ethanol content of the mixed fermentation significantly reduced ($p < 0.05$), and the ethanol reduction could reach 0.24 to 0.71% vol. Whether the *H. uvarum* or *P. fermentans* is involved in the fermentation mode, the same pattern can be found that the 10:1 inoculation fermentation method (including sequential and simultaneous inoculation) can reduce more ethanol than the 1:1 low ratio inoculation fermentation method. In addition, under the same yeast of inoculation method, sequential inoculation reduced more alcohol than simultaneous inoculation. The titratable acidity content of grape juice initially was 5.4 g/L, and after the fermentation, the titratable acidity decreased, which may be the reason for the metabolism by yeast. In addition, it can be found that the titratable acidity content of all the mixed fermentation schemes was not significantly different from the CK ($p < 0.05$). Glycerol is often considered the most abundant compound besides water and alcohol, usually at 7 to 10 g/L in dry wine, which increases the body and texture of the wine [22]. Compared with the CK, the glycerol content of the mixed fermentation group was significantly higher than the CK. The sequential inoculation of *P. fermentans* (WSP1, WSP10) could especially increase the glycerol content by 0.43 and 0.47 g/L. Furthermore, it can be found that inoculation time had little effect on glycerol content, and the yield of glycerol also increased with the number of vaccinated NS. Volatile acid is one of the typical products of alcoholic fermentation, and acetic acid is the most common volatile acid in wine, which can result in bad smells, such as vinegar and spicy, to wine when the content is too high; it is considered as microbially broken (including acetic acid bacteria or lactic acid bacteria), thus affecting the quality of wine. As shown in Table 4, all mixed fermentation can increase the volatile acid content from 0.01 to 0.38 g/L compared to the CK. The sequential inoculation of *H. uvarum* produced more volatile acids, but the content of volatile acids in all wine samples was less than 1.2 g/L (measured by acetic acid), which met the Chinese national standards.

*3.4. Volatile Aroma Compounds after Alcoholic Fermentation*

Volatile aroma substances have volatile properties and can produce certain odors. They can be divided into three categories: variety aroma (class I aroma), fermented aroma (class II aroma), and aged aroma (class III aroma) [31]. At present, more than 1000 volatile substances have been detected in wine, including higher alcohols, esters, phenol, terpenes, pyrazines, and sulfur compounds [31,32]. The species, concentration, sensory thresholds, and interactions of these compounds together determine the aroma quality of the wine [32]. In general, volatile aroma substances are one of the main factors affecting the sensory quality and typicality of wine. Different yeast and inoculation techniques can also affect the aroma composition of wine.

The volatile aroma composition and content of each fermentation group were determined after the fermentation of Italian Riesling, and the measurement results are shown in Table 5. A total of 46 volatile aroma substances were detected in nine groups of fermented wine samples, including the most esters (28), second higher alcohols (6), five types of variety aroma substances, three fatty acid substances, and four other compounds. The total amount of these volatile aromatic substances in the wine is between 446.71 and 653.61 mg/L. Except for WCP1, the total amount of volatile substances in the mixed fermentation group was significantly higher than that of CK. Among these, the total amount of volatile substances was the highest under the WSP10 inoculation method, increasing by 46% compared to CK. The total volatile aroma of *H. uvarum* was not affected by inoculation time and inoculation amount and the content difference was not significant ($p < 0.05$).

Variety aroma is usually directly derived from the grape itself according to its metabolic pathway; they can be divided into six categories, including terpenes, C6 compounds, C13-norisoprenoids, volatile phenols, pyrazines, and thiols [33,34]. These compounds do not exist as separate individuals but interact with each other collectively to increase the wine flavor. The variety aroma of the wine is made up of these compounds. They are mg/L in wine; the low is only ng/L or even lower [33]. There were five varieties measured in the wine samples in a range of 0.96 to 1.98 mg/L, including one terpene (linalool),

one C13-norisoprenoids (*β*-damasone), one C6 alcohol (1-hexanol), one volatile phenol (2,4-tert-butyl phenol), and one thiol (tert-hexadothiol). From Table 5, we can find thiols only appear in the fermentation of *H. uvarum*. In addition, all mixed fermentation protocols can significantly increase the content of wine linalool. For the total amount of variety aroma, all mixed cultures except WSP1 and WSH1 were significantly higher than CK ($p < 0.05$). Fermented aroma refers to the aroma substances produced in the alcoholic fermentation process of wine, which plays an important position in wine.

Higher alcohols are formed in the secondary products of alcoholic fermentation, are the product of amino acid or sugar metabolism of yeast during alcoholic fermentation, are high in content and can be recognized by their strong pungent odor and taste. The higher alcohol ranged between 164.63 and 239.58 mg/L, and the highest levels of higher alcohols were isopentol and phenethyl alcohol. Only WCP1 had less higher alcohol content than CK, and all other mixed fermentation methods were higher than CK. In particular, the WSP10 content is at the highest level. Nonanol can bring green and orange aroma to wine, and according to Table 5, the CK has the highest content, which is significantly higher than other fermentation groups except WCH1. In addition, it can be found that *H. uvarum* can increase the aroma of 1-dodecanol and then increase the fragrance of the wine.

Esters can bring a fruity aroma to wine. The test found the esters in the wine include acetate esters, alcohol esters, and other esters, with a total content between 230.42 and 360.45 mg/L. The main ester substances are ethyl ester, ethyl decanoate, and isoamyl acetate, with fruit aromas such as pineapple, pear, and fresh banana, as shown in Table 5. In addition, all mixed fermentation can increase the total amount of esters, including the fermentation of *P. fermentans*; the 10:1 inoculation fermentation mode WSP10 and WCP10 can especially increase the total amount of esters by 49.4% and 56.5%, respectively, compared with the control. The highest esters shown in Table 5 were ethyl esters, which are more abundant in *P. fermentans*. Acetate esters are abundant in *H. uvarum*.

To better distinguish the differences between treatments using R software for analysis, the cluster heatmap results are shown in Figure 3. Nine treatments of fermented wine samples are divided into three fermentation groups. The first includes WSP10, WSP1, and WCP10, which are inoculated with *P. fermentans*. The second type is inoculated with *H. uvarum*, and the third category is CK and WCP1. Using longitudinal analysis, the volatile aroma substances can be divided into three categories. The first category of aroma components includes isobutanol, ethyl acetate, ethyl acetate, and isoamyl acetate, including five kinds of acetate substances, which are abundant due to the fermentation method of adding *H. uvarum*. The second type of aroma components includes nonanal, ethyl laurate, methyl caprylate, and ethyl nonylate, styrene, totaling 23 substances, including seven ethanol esters, which are the most abundant in the sequential inoculation in the fermentation of *P. fermentans* and 10:1 simultaneous inoculation. The third class of aroma substances includes 12 substances, such as 1-hexanol, ethyl succinate, *β*-damasone, and caryophyllene, which are irregular and less abundant in CK, while they are present in the fermentation of both *P. fermentans* and *H. uvarum*.

The concentrations of aroma compounds from higher to lower are expressed in red, white, and blue, respectively, which are only compared in the same rows and not the same columns.

**Table 5.** Volatile compounds of the wines obtained by different fermentation treatments/(mg/L).

| Number | Compound | Aroma Substance Concentration (mg/L) | | | | | | | | | The Threshold Value (mg/L) | Aroma Description [17,18,20,34–41] |
|---|---|---|---|---|---|---|---|---|---|---|---|---|
| | | CK | WSH1 | WSH10 | WCH1 | WCH10 | WSP1 | WSP10 | WCP1 | WCP10 | | |
| | Variety aroma | | | | | | | | | | | |
| A1 | 1-Hexanol | 0.67 ± 0.04 d | 0.68 ± 0.14 d | 0.69 ± 0.05 d | 0.87 ± 0.01 abc | 0.96 ± 0.04 a | 0.82 ± 0.07 abcd | 0.73 ± 0.01 cd | 0.77 ± 0.03 bcd | 0.9 ± 0.08 ab | 8 [18] | Flowers, fruity, green grass |
| A2 | Tert-hexadothiol | 0 ± 0 e | 0.01 ± 0 c | 0.02 ± 0 a | 0.01 ± 0 d | 0.02 ± 0 b | 0 ± 0 e | 0 ± 0 e | 0 ± 0 e | 0 ± 0 e | NF | NF |
| A3 | β-damasone | 0.23 ± 0.01 d | 0.23 ± 0.01 d | 0.53 ± 0.01 b | 0.41 ± 0.04 c | 0.3 ± 0.07 d | 0.22 ± 0.05 d | 0.5 ± 0.02 b | 0.53 ± 0.02 b | 0.74 ± 0.02 a | 0.05 [19] | Sweet apples, plums, and canned peaches |
| A4 | 2,4-di-tert-butylphenol | 0.01 ± 0 b | 0.22 ± 0.02 ab | 0.53 ± 0.38 a | 0.64 ± 0.33 a | 0.38 ± 0.04 ab | 0.2 ± 0.06 ab | 0.28 ± 0.07 ab | 0.34 ± 0 ab | 0.26 ± 0.15 ab | 0.2 [34] | NF |
| A5 | Linalool | 0.05 ± 0.01 e | 0.06 ± 0.01 d | 0.07 ± 0 abc | 0.06 ± 0 d | 0.07 ± 0 ab | 0.06 ± 0 cd | 0.07 ± 0 a | 0.07 ± 0 bc | 0.07 ± 0 a | 0.025 [35] | Flowers, musk, lavender incense |
| | Total | 0.96 ± 0.05 d | 1.2 ± 0.15 cd | 1.84 ± 0.42 ab | 1.98 ± 0.36 a | 1.73 ± 0.15 ab | 1.31 ± 0.07 bcd | 1.58 ± 0.08 abc | 1.71 ± 0.01 abc | 1.98 ± 0.21 a | | |
| | Fermented aroma | | | | | | | | | | | |
| | Higher alcohols | 182.85 ± 5.89 d | 201.66 ± 4 bc | 198 ± 5.33 bcd | 203.76 ± 13.28 bc | 190.24 ± 10.55 cd | 209.03 ± 0.13 b | 239.58 ± 3.76 a | 164.62 ± 9.89 e | 200.69 ± 4.53 bcd | | |
| B1 | Isobutanol | 0.53 ± 0.02 e | 0.72 ± 0.04 c | 0.75 ± 0.03 bc | 0.81 ± 0.03 ab | 0.76 ± 0.03 bc | 0.61 ± 0.01 d | 0.87 ± 0 a | 0.77 ± 0.01 bc | 0.81 ± 0.05 ab | 40 [36] | Fusel, alcohol, light sweet |
| B2 | Isoamylol | 153.71 ± 5.62 b | 161.11 ± 1.48 ab | 158.91 ± 5.13 ab | 158.28 ± 10.98 ab | 155.16 ± 12.32 ab | 165.01 ± 3.22 ab | 170.21 ± 3.08 a | 118.41 ± 4.13 c | 159.49 ± 4.11 ab | 30 [19] | Alcohol taste, bitter taste, and nail polish |
| B3 | 2,3-Butanediol | 0.24 ± 0.01 e | 0.29 ± 0.01 e | 0.41 ± 0.03 d | 0.42 ± 0.01 cd | 0.74 ± 0 a | 0.49 ± 0.05 c | 0.6 ± 0.05 b | 0.48 ± 0 cd | 0.72 ± 0.05 a | 120 [36] | Rubber, cream |
| B4 | Phenethyl alcohol | 27.46 ± 0.28 e | 38.53 ± 2.35 bc | 36.91 ± 0.31 cd | 43.13 ± 2.35 bc | 32.41 ± 1.89 de | 42.06 ± 3.07 bc | 66.84 ± 0.71 a | 44.06 ± 5.75 b | 38.81 ± 0.38 bc | 14 [17] | Honey, rose, lilacs |
| B5 | Dodecanol | 0.58 ± 0.01 c | 0.82 ± 0.18 ab | 0.89 ± 0.05 a | 0.82 ± 0 ab | 0.97 ± 0.07 a | 0.67 ± 0.07 bc | 0.86 ± 0.04 a | 0.64 ± 0.01 c | 0.68 ± 0.06 bc | 1 [17] | Floral |
| B6 | Nonanol | 0.32 ± 0 a | 0.19 ± 0.02 c | 0.13 ± 0 d | 0.29 ± 0.01 ab | 0.2 ± 0.02 b | 0.19 ± 0 c | 0.2 ± 0.01 c | 0.27 ± 0.01 b | 0.19 ± 0.01 c | 0.6 [34] | Herbaceous, roses, oranges |
| | Ester | 230.42 ± 3.46 e | 272.48 ± 20.63 cd | 271.23 ± 6.92 cd | 276.4 ± 0.91 c | 269.09 ± 3.44 cd | 321.45 ± 0.53 b | 344.28 ± 3.31 a | 257.15 ± 5.58 d | 360.45 ± 1.08 a | | |
| | Acetate ester | 45.44 ± 1.41 de | 55.37 ± 1.26 ab | 55.1 ± 0.46 ab | 49.34 ± 1.46 cd | 55.22 ± 0.24 ab | 45.04 ± 5.04 de | 56.75 ± 0.02 a | 42.26 ± 0.71 e | 51.11 ± 2.1 bc | | |
| C1 | Ethyl acetate | 5.65 ± 0.08 c | 8.26 ± 0.07 a | 8.37 ± 0.16 a | 8.06 ± 0 a | 8.23 ± 0.07 a | 6.6 ± 0.3 b | 7.05 ± 0.27 b | 5.31 ± 0.06 c | 6.73 ± 0.36 b | 7.5 [26] | Fruity, sweet taste, and nail polish |
| C2 | Isobutyl acetate | 0.16 ± 0.01 cd | 0.21 ± 0 abc | 0.22 ± 0.02 ab | 0.2 ± 0.01 abc | 0.2 ± 0 abc | 0.17 ± 0.05 bcd | 0.13 ± 0.02 d | 0.13 ± 0.01 d | 0.23 ± 0.02 a | 1.6Aroma of Six | Fruity, pear, banana |
| C3 | Isoamyl acetate | 23.49 ± 2.24 ab | 27.11 ± 0.26 a | 26.93 ± 0.27 a | 23.73 ± 0.69 ab | 26.05 ± 0.41 a | 19 ± 4.52 c | 24.46 ± 0.74 ab | 21.13 ± 0.11 bc | 23.73 ± 1.22 ab | 0.03 [19] | Fruity, fresh banana |
| C4 | Hexyl acetate | 6.44 ± 0.21 c | 8.13 ± 0.19 a | 8.35 ± 0.08 a | 7.25 ± 0.27 b | 8.1 ± 0.02 a | 6.74 ± 0.41 bc | 8.08 ± 0.6 a | 5.67 ± 0.07 d | 7.93 ± 0.34 a | 1.5 [17] | Fruit fragrance, pear, cherry |
| C5 | Heptyl acetate | 0.07 ± 0.01 c | 0.12 ± 0.01 a | 0.13 ± 0.02 a | 0.1 ± 0.01 b | 0.12 ± 0 a | 0.05 ± 0 cd | 0.11 ± 0.01 ab | 0.04 ± 0 d | 0.04 ± 0.01 d | NF | Cherry, pear |
| C6 | Octyl acetate | 0.15 ± 0 cd | 0.18 ± 0.01 b | 0.18 ± 0 b | 0.11 ± 0.01 ef | 0.11 ± 0.01 ef | 0.13 ± 0.01 de | 0.22 ± 0.02 a | 0.1 ± 0.02 f | 0.17 ± 0.01 bc | NF | NF |
| C7 | Phenylethyl acetate | 9.48 ± 0.71 d | 11.35 ± 1.24 bc | 10.91 ± 0.04 bcd | 9.89 ± 1.01 cd | 12.41 ± 0.61 b | 12.35 ± 0.25 b | 16.7 ± 1.09 a | 9.87 ± 0.43 cd | 12.28 ± 0.19 b | 1.5 [17] | Floral, wood |
| | Ethyl ester | 173.7 ± 2.03 d | 204.04 ± 17.44 c | 202.16 ± 6.17 c | 211.28 ± 1.2 c | 203.94 ± 3.43 c | 258.65 ± 4.69 b | 264.98 ± 3.27 b | 201.04 ± 4.43 c | 290.62 ± 0.28 a | | |
| D1 | Ethyl butyrate | 0.5 ± 0.02 c | 0.66 ± 0.04 b | 0.6 ± 0.02 bc | 0.64 ± 0.03 b | 0.83 ± 0.01 a | 0.49 ± 0.04 c | 0.66 ± 0.11 b | 0.64 ± 0.02 b | 0.68 ± 0.05 b | 0.02 [17] | Strawberries, apples, bananas |
| D2 | Ethyl hexylate | 35.37 ± 0.64 d | 43.11 ± 3.17 abc | 38.47 ± 0.68 cd | 46.46 ± 1.34 a | 45.91 ± 1.71 a | 40.18 ± 2.39 bcd | 43.9 ± 3.88 ab | 41.99 ± 0.93 abc | 44.75 ± 1.75 ab | 0.014 [18] | Green apple, fruity, strawberry |
| D3 | Ethyl oenanthate | 0.07 ± 0.01 d | 0.07 ± 0.01 cd | 0.08 ± 0.01 bcd | 0.1 ± 0 b | 0.12 ± 0.02 a | 0.08 ± 0 bcd | 0.09 ± 0.01 ab | 0.08 ± 0.01 bcd | 0.07 ± 0 d | 0.22 [17] | Pineapple, fruity |
| D4 | Ethyl caprylate | 44.69 ± 1.01 cd | 50.37 ± 8.63 bc | 45.43 ± 2.53 cd | 45 ± 0.61 cd | 43.08 ± 0.72 cd | 54.5 ± 1.43 ab | 59.2 ± 1.09 a | 37.73 ± 1.67 d | 59.79 ± 1.37 a | 0.005 [39] | Pineapple, pear, floral |
| D5 | Ethyl pelargonate | 0.35 ± 0.02 f | 0.44 ± 0.07 de | 0.39 ± 0.03 ef | 0.59 ± 0.04 bc | 0.5 ± 0.03 cd | 0.6 ± 0.01 b | 0.74 ± 0.03 a | 0.31 ± 0.04 f | 0.51 ± 0.03 bcd | 1. 2 [38] | Fruity, rose, wax, rum |
| D6 | Ethyl caprate | 45.4 ± 1.26 b | 46.06 ± 3.75 b | 47.92 ± 2.83 b | 45.6 ± 0.22 b | 43.19 ± 0.06 b | 54.97 ± 0.99 a | 58.72 ± 0.84 a | 37.08 ± 0.99 c | 58.98 ± 2.85 a | 0.2 [19] | Fruity, fat, wax |

**Table 5.** *Cont.*

| Number | Compound | Aroma Substance Concentration (mg/L) | | | | | | | | | The Threshold Value (mg/L) | Aroma Description [17,18,20,34–41] |
|---|---|---|---|---|---|---|---|---|---|---|---|---|
| | | CK | WSH1 | WSH10 | WCH1 | WCH10 | WSP1 | WSP10 | WCP1 | WCP10 | | |
| D7 | Ethyl succinate | 0.26 ± 0 f | 0.25 ± 0.02 f | 0.41 ± 0 e | 0.41 ± 0.04 e | 0.6 ± 0.01 a | 0.54 ± 0.02 b | 0.43 ± 0.01 de | 0.5 ± 0.05 bc | 0.47 ± 0.01 cd | 0.1 [20] | NF |
| D8 | Ethyl 9-decenoate | 20.85 ± 1.15 d | 28.13 ± 0.79 a | 27.48 ± 0.28 ab | 27.04 ± 0.87 ab | 26.84 ± 1.12 ab | 22.6 ± 2.52 cd | 28.72 ± 2.05 a | 27.02 ± 0.02 ab | 24.72 ± 0.63 bc | 0.1 [34] | Fruity |
| D9 | Ethyl laurate | 25.89 ± 0.45 g | 34.31 ± 2.48 f | 40.7 ± 0.36 ef | 44.47 ± 1.31 e | 42.45 ± 3.4 e | 82.74 ± 7.22 b | 70.55 ± 0.79 c | 54.28 ± 0.45 d | 99.03 ± 1.27 a | 1.5 [35] | Sweet, floral, fruitt, creamy |
| D10 | Ethyl myristate | 0.22 ± 0.04 de | 0.44 ± 0.06 cde | 0.52 ± 0.05 cde | 0.62 ± 0.02 bcde | 0.14 ± 0.02 e | 1.1 ± 0 ab | 1.2 ± 0.04 a | 0.72 ± 0.1 abcd | 0.91 ± 0.62 abc | NF | NF |
| D11 | Ethyl palmitate | 0.1 ± 0.01 c | 0.2 ± 0.09 bc | 0.18 ± 0.01 bc | 0.36 ± 0.04 b | 0.26 ± 0.02 bc | 0.86 ± 0.01 a | 0.78 ± 0.06 a | 0.69 ± 0.24 a | 0.73 ± 0.16 a | 1.5 [34] | NF |
| | Other esters | 11.28 ± 0.02 e | 13.07 ± 1.93 d | 13.97 ± 0.29 d | 15.78 ± 0.65 c | 9.93 ± 0.23 e | 17.77 ± 0.18 b | 22.55 ± 0.01 a | 13.84 ± 0.44 d | 18.72 ± 0.74 b | | |
| E1 | Methyl caproate | 0.07 ± 0.01 bc | 0.09 ± 0.02 b | 0.07 ± 0.01 bc | 0.08 ± 0.02 b | 0.05 ± 0 c | 0.07 ± 0.01 bc | 0.09 ± 0.02 b | 0.13 ± 0 a | 0.09 ± 0 b | NF | NF |
| E2 | Methyl caprylate | 2.4 ± 0.03 de | 2.48 ± 0.36 cde | 2.47 ± 0.09 cde | 2.94 ± 0 b | 0.5 ± 0.02 f | 2.66 ± 0.18 bcd | 3.66 ± 0.22 a | 2.09 ± 0.05 e | 2.84 ± 0.12 bc | 0.2 [38] | Citrus |
| E3 | Amyl hexylate | 0.87 ± 0.01 de | 1.05 ± 0.14 bcd | 0.84 ± 0.04 e | 1.15 ± 0.12 b | 0.81 ± 0.06 e | 1.07 ± 0.02 bc | 1.42 ± 0.05 a | 0.91 ± 0.02 cde | 1.13 ± 0.15 b | NF | Apple, pineapple |
| E4 | Propyl octanoate | 0.33 ± 0.01 cd | 0.54 ± 0.1 ab | 0.37 ± 0.04 bcd | 0.46 ± 0.01 bc | 0.24 ± 0.07 d | 0.53 ± 0.01 ab | 0.69 ± 0.09 a | 0.25 ± 0.04 d | 0.49 ± 0.14 bc | NF | NF |
| E5 | Isobutyl caprylate | 0.44 ± 0.01 d | 0.47 ± 0.09 cd | 0.51 ± 0.01 cd | 0.51 ± 0.04 cd | 0.49 ± 0.05 cd | 0.63 ± 0.01 b | 0.8 ± 0.05 a | 0.55 ± 0.01 bc | 0.84 ± 0.02 a | NF | NF |
| E6 | Methyl caprate | 3.09 ± 0.02 b | 3.38 ± 0.58 b | 3.44 ± 0.2 b | 3.14 ± 0.02 b | 3.49 ± 0.27 b | 5.1 ± 0.11 a | 5.32 ± 0.01 a | 3.54 ± 0.2 b | 5.08 ± 0.08 a | NF | Wine |
| E7 | Isoamyl caprylate | 3 ± 0.04 e | 3.55 ± 0.41 cd | 3.94 ± 0.22 c | 4.78 ± 0.31 b | 3.3 ± 0.07 de | 4.67 ± 0.12 b | 6.95 ± 0.16 a | 3.49 ± 0.03 cd | 4.72 ± 0.14 b | 0.125 [40] | Fruity, cheese |
| E8 | n-Capric acid isobutyl ester | 0.21 ± 0 e | 0.29 ± 0.03 d | 0.39 ± 0 c | 0.36 ± 0.02 c | 0.22 ± 0.07 e | 0.5 ± 0.01 b | 0.56 ± 0.01 ab | 0.37 ± 0 c | 0.61 ± 0.04 a | NF | NF |
| E9 | Methyl laurate | 0.18 ± 0.01 f | 0.29 ± 0.01 de | 0.38 ± 0 c | 0.36 ± 0.07 cd | 0.23 ± 0.02 ef | 0.35 ± 0.06 cd | 0.35 ± 0.02 cd | 0.46 ± 0.01 b | 0.75 ± 0 a | NF | NF |
| E10 | Isoamyl caprate | 0.68 ± 0.07 c | 0.93 ± 0.2 c | 1.56 ± 0.23 b | 2.01 ± 0.12 b | 0.61 ± 0.2 c | 2.18 ± 0.16 ab | 2.71 ± 0.01 a | 2.05 ± 0.18 b | 2.18 ± 0.64 ab | NF | NF |
| | Fatty acid | 31.74 ± 1.75 g | 35.79 ± 3.01 fg | 38.29 ± 0.34 efg | 45.23 ± 3.23 cde | 40.64 ± 1.22 def | 51.06 ± 2.96 bc | 67.07 ± 1.01 a | 52.88 ± 2.68 b | 47.19 ± 6.12 bcd | | |
| F1 | Hexanoic acid | 1.61 ± 0.09 a | 1.52 ± 0.07 ab | 1.27 ± 0.02 cd | 1.59 ± 0.18 a | 1.33 ± 0.08 bcd | 1.28 ± 0.08 cd | 1.46 ± 0.07 abc | 1.17 ± 0.04 d | 1.67 ± 0.02 a | 0.42 [19] | Cheese, foliage |
| F2 | Octanoic acid | 18.21 ± 0.58 e | 23.47 ± 2.23 d | 24.12 ± 0.35 d | 29.22 ± 3.53 bc | 27.55 ± 0.26 cd | 32.89 ± 1.55 b | 46.71 ± 0.42 a | 33.04 ± 2.88 b | 26.44 ± 1.16 cd | 0.5 [17] | Butter, almond |
| F3 | n-Capric acid | 11.91 ± 1.09 d | 10.79 ± 0.71 d | 12.91 ± 0.72 cd | 14.42 ± 0.48 bcd | 11.76 ± 1.56 d | 16.89 ± 1.33 abc | 18.9 ± 0.66 ab | 18.68 ± 0.16 ab | 19.08 ± 4.98 a | 1 [35] | Caramel, milk, fat, rotten |
| | Others | 0.74 ± 0.04 c | 0.75 ± 0.02 c | 0.69 ± 0.01 de | 0.71 ± 0.02 cd | 0.64 ± 0.01 ef | 0.62 ± 0 f | 1.09 ± 0.03 b | 1.18 ± 0.03 a | 1.07 ± 0 b | | |
| G1 | Styrene | 0.13 ± 0.01 ab | 0.13 ± 0.01 ab | 0.11 ± 0.01 bc | 0.09 ± 0.01 c | 0.07 ± 0.01 c | 0.08 ± 0.01 c | 0.13 ± 0.02 ab | 0.1 ± 0.02 c | 0.15 ± 0.01 a | 0.73 [40] | NF |
| G2 | Decanal | 0.14 ± 0.01 b | 0.14 ± 0.03 b | 0.15 ± 0.01 b | 0.06 ± 0.01 c | 0.15 ± 0.07 b | 0.1 ± 0.02 bc | 0.14 ± 0 b | 0.25 ± 0.02 a | 0.28 ± 0.02 a | 0.01 [37] | NF |
| G3 | Caryophyllene | 0.2 ± 0.03 cd | 0.28 ± 0.03 bc | 0.26 ± 0.01 bcd | 0.32 ± 0.03 b | 0.22 ± 0.06 cd | 0.17 ± 0.04 d | 0.48 ± 0.05 a | 0.51 ± 0.06 a | 0.24 ± 0 bcd | NF | NF |
| G4 | Nonanal | 0.27 ± 0.02 c | 0.2 ± 0.01 e | 0.18 ± 0 e | 0.24 ± 0.01 d | 0.2 ± 0 e | 0.28 ± 0.01 c | 0.34 ± 0.01 b | 0.33 ± 0.03 b | 0.4 ± 0.01 a | 0.015 [26] | Herbaceous |
| | Total | 445.75 ± 4.14 e | 510.67 ± 27.66 cd | 508.21 ± 11.92 cd | 526.1 ± 17.41 c | 500.61 ± 15.2 cd | 582.17 ± 2.3 b | 652.03 ± 8.11 a | 475.84 ± 18.11 de | 609.4 ± 9.56 b | | |
| | Sum | 446.71 ± 4.19 e | 511.88 ± 27.81 cd | 510.04 ± 11.5 cd | 528.09 ± 17.04 c | 502.35 ± 15.35 cd | 583.48 ± 2.23 b | 653.61 ± 8.19 a | 477.55 ± 18.12 de | 611.38 ± 9.77 b | | |

Note: The data are presented in the form of 'mean ± standard deviation'. In the rows, different lowercases represent significant differences between treatments (Duncan's test, $p < 0.05$). 'NF' is not found.

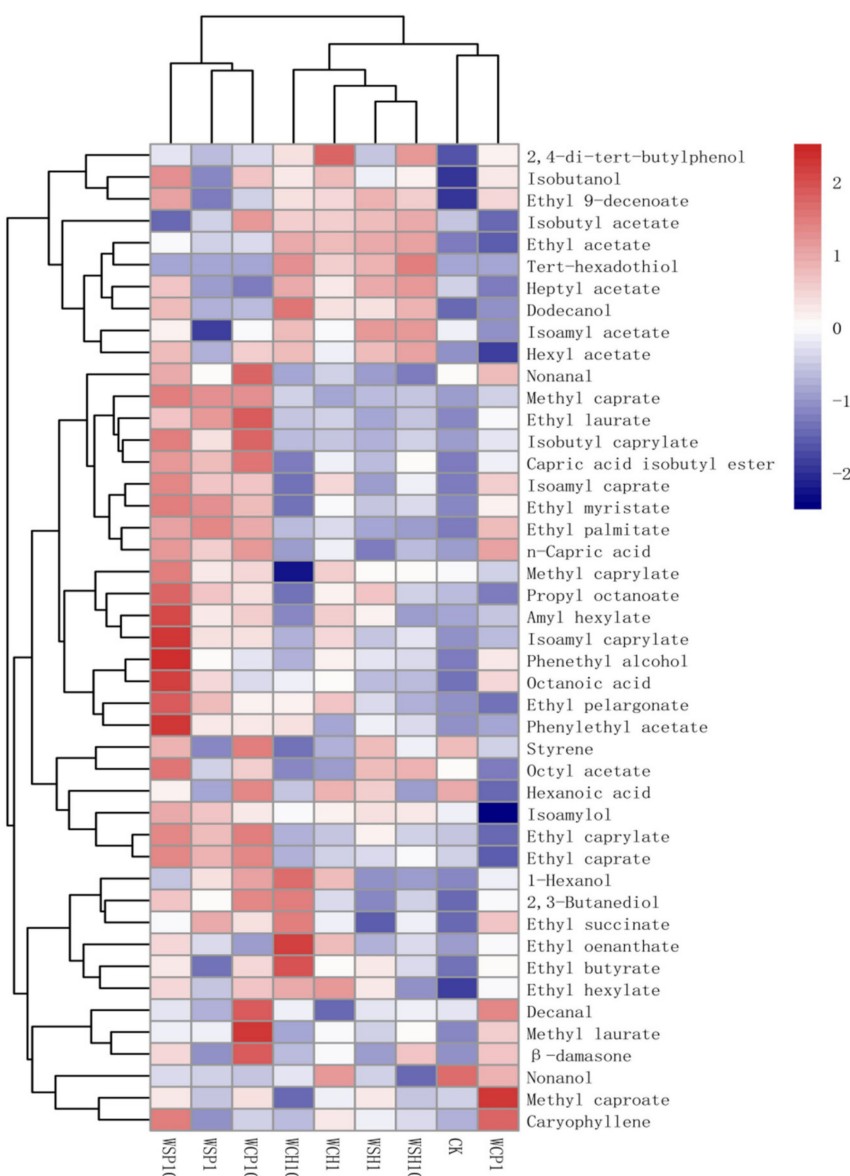

**Figure 3.** Cluster heat map analysis of aroma components of wine samples.

*3.5. Sensory Evaluation of Wine*

After the fermentation, the sensory test of the wine was carried outbased on the hundred percent sensory evaluation system formulated by the International Organisation of Vine and Wine, which made a comprehensive evaluation of the color, aroma, and taste of the wine. As shown in Table 6, the sensory evaluation scores of each wine sample ranged from 81.1 to 89.3. In terms of chroma, hue, and clarity, the difference between each wine sample was not significant, and they were all clear, bright, and glossy wines. In the evaluation of aroma purity, WSP10 and WCP10 were scored significantly above CK. WSP10 was scored significantly higher than CK in terms of aroma elegance and harmony ($p < 0.05$). In addition, it can be found that a 10:1 fermentation method of *H. uvarum* is not ideal in aroma score, because its aroma elegance is significantly lower than CK and other fermentation groups, and WSH10 is not inferior to CK in terms of aroma purity and body. The difference between the groups was not significant in terms of most taste (body, structure, harmony, aroma persistence, and finish), while all mixed fermentation groups were stronger than CK in terms of taste purity. Overall, the taster scored higher on the wine produced by *P. fermentans*, with both WSP10 and WCP10 significantly higher than the CK. WSH10 has a defective odor (oxidation and yeast flavor), resulting in the lowest score.

Therefore, a low proportion of *H. uvarum* can increase the aromas of floral, mint, and roast toast, while excessive inoculation may increase the risk of wine oxidation, as explained by the previous volatile acid content. A high proportion of *P. fermentans* can increase the aroma of lemon, cream, and almond and improve the complexity of wine.

**Table 6.** Sensory quality score and sensory characteristics of wine samples.

| Sensory Features | | Wine Samples | | | | | | | | |
|---|---|---|---|---|---|---|---|---|---|---|
| | | CK | WSH1 | WSH10 | WCH1 | WCH10 | WSP1 | WSP10 | WCP1 | WCP10 |
| Color | Clarity | 5.4 ± 0.84 a | 5.8 ± 0.42 a | 5.2 ± 0.79 a | 5.6 ± 0.7 a | 5.6 ± 0.52 a | 5.7 ± 0.48 a | 5.8 ± 0.42 a | 5.6 ± 0.7 a | 5.7 ± 0.48 a |
| | Chroma | 5.4 ± 0.7 a | 5.8 ± 0.42 a | 5.8 ± 0.42 a | 5.5 ± 0.53 a | 5.6 ± 0.52 a | 5.6 ± 0.52 a | 5.6 ± 0.52 a | 5.5 ± 0.53 a | 5.4 ± 0.52 a |
| | Hue | 5.2 ± 0.92 b | 5.8 ± 0.42 a | 5.4 ± 0.52 ab | 5.6 ± 0.52 ab | 5.5 ± 0.53 ab | 5.6 ± 0.52 ab | 5.4 ± 0.52 ab | 5.5 ± 0.53 ab | 5.8 ± 0.42 a |
| Aroma | Pure Degree | 4.8 ± 0.79 c | 5.2 ± 0.63 abc | 4.2 ± 0.79 d | 5.2 ± 0.42 abc | 4.9 ± 0.32 c | 5.1 ± 0.32 bc | 5.7 ± 0.67 a | 5 ± 0.47 c | 5.6 ± 0.52 ab |
| | Wine Body | 6.8 ± 0.79 ab | 7.1 ± 0.74 ab | 6.1 ± 0.57 c | 6.6 ± 0.7 bc | 6.8 ± 0.79 ab | 6.8 ± 0.63 ab | 7.4 ± 0.52 a | 7.1 ± 0.57 ab | 6.9 ± 0.74 ab |
| | Elegant | 6.8 ± 0.79 b | 6.8 ± 0.42 b | 5.9 ± 0.74 c | 6.8 ± 0.42 b | 5.6 ± 0.52 c | 7.2 ± 0.63 ab | 7.5 ± 0.53 a | 6.7 ± 0.48 b | 7.2 ± 0.63 ab |
| | Harmony | 6.4 ± 0.7 bc | 6.8 ± 0.63 b | 6 ± 0.67 c | 6.8 ± 0.42 b | 6.9 ± 0.74 b | 7 ± 0.67 ab | 7.5 ± 0.53 a | 6.8 ± 0.63 b | 6.8 ± 0.42 b |
| Taste | Pure Degree | 4.7 ± 0.48 b | 5.3 ± 0.67 a | 5 ± 0.47 ab | 4.8 ± 0.79 ab | 5 ± 0.47 ab | 5.3 ± 0.48 a | 5.1 ± 0.57 ab | 4.9 ± 0.57 ab | 5.3 ± 0.48 a |
| | Wine Body | 6.3 ± 0.82 a | 7 ± 0.47 a | 6.8 ± 0.79 a | 7 ± 0.82 a | 6.7 ± 0.67 a | 6.9 ± 0.74 a | 6.8 ± 0.79 a | 6.9 ± 0.57 a | 6.9 ± 0.57 a |
| | Structure | 6.4 ± 0.84 a | 6.4 ± 0.52 a | 6.7 ± 0.82 a | 6.7 ± 0.67 a | 6.8 ± 0.79 a | 6.8 ± 0.63 a | 6.9 ± 0.74 a | 6.6 ± 0.7 a | 6.8 ± 0.63 a |
| | Harmony | 6.4 ± 0.84 a | 6.4 ± 0.52 a | 6.5 ± 0.71 a | 6.5 ± 0.85 a | 6.7 ± 0.48 a | 6.7 ± 0.67 a | 7 ± 0.67 a | 6.6 ± 0.7 a | 7 ± 0.47 a |
| | Persistence | 6.6 ± 0.7 a | 6.4 ± 0.7 a | 6.3 ± 0.95 a | 6.6 ± 0.84 a | 6.7 ± 0.67 a | 6.6 ± 0.52 a | 6.6 ± 0.7 a | 6.7 ± 0.67 a | 6.6 ± 0.7 a |
| | Finish | 4.8 ± 0.79 a | 4.6 ± 0.52 a | 4.4 ± 0.84 a | 4.8 ± 0.42 a | 4.8 ± 0.63 a | 4.7 ± 0.48 a | 4.9 ± 0.57 a | 4.5 ± 0.53 a | 5 ± 0.47 a |
| | Comprehensive evaluation | 6.5 ± 0.53 ab | 6.8 ± 0.63 ab | 6.8 ± 0.63 ab | 6.4 ± 0.7 b | 6.6 ± 0.7 ab | 6.9 ± 0.57 ab | 7.1 ± 0.74 a | 6.7 ± 0.48 ab | 7.1 ± 0.32 a |
| Total points | | 82.5 ± 6.9 cd | 86.2 ± 3.05 abc | 81.1 ± 5.8 d | 84.9 ± 3.73 abcd | 84.2 ± 5.05 bcd | 86.9 ± 3.31 abc | 89.3 ± 3.06 a | 85.1 ± 4.25 abcd | 88.1 ± 2.92 ab |
| Aromatic features | | Melon, Fruit stem, Alcohol | Flowers, Mint, Roast toast | Yeast cell, Flowers, Oxidation | Floral, Alcohol | Floral, mint Mild oxidation | Lemon, Mint, Grapefruit | Lemon, Mint, Cream | Melo, Cream, Mint | Lemon, Mint, Almond |

Note: The data are presented in the form of 'mean ± standard deviation'. In the rows, different lowercases represent significant differences between treatments (Duncan's test, $p < 0.05$).

## 4. Discussion

Our work observed the interactions between *S. cerevisiae* and NS strains during mixed fermentations. *S. cerevisiae* was the dominant species driving the AF process and had some inhibitory effects on the growth of NS yeasts, which can be evidenced by the shorter survival time of NS cells in simultaneous inoculums compared to sequential inoculations (Figure 2). Studies have suggested that the development of NS yeasts can be impeded or stagnated by a few metabolites produced by *S. cerevisiae*, such as ethanol and medium-chain fatty acids [42]. The killer toxins released by *S. cerevisiae*, including glucanases, protein toxoids, and anti-microbial peptides, were found to be death-inducing factors for *H. uvarum* [43]. Other mechanisms, such as quorum sensing and cell–cell interactions were also reported to result in the early death of *L. thermotolerans* and *T. delbrueckii* during co-fermentations with *S. cerevisiae* [44]. Additionally, most NS yeasts have lower adaptability to the fermentation environment (oxidation, high temperature, sulfur dioxide, ethanol, etc.), leading to their weak competitiveness against *Saccharomyces* spp. [45,46].

Despite this, growth inhibition happens reciprocally and is not one-sided. We discovered that mixed fermentations had lower cell densities of *S. cerevisiae* than pure-inoculated environments. Higher NS-yeast cells caused a more noticeable suppression, especially when they were introduced 48 h before *S. cerevisiae*. These findings are consistent with those reported by Domizio and colleagues [47]. This might be explained by the various yeast species present in the fermentation mixture having different rates of nutrient assimilation. A paucity of required nitrogen sources and constrained cell proliferation for *S. cerevisiae* may come from the early inoculation of NS yeasts, which consume some of the yeast assimilable nitrogen [43,48].

The NS yeasts are generally constrained in their ability to produce ethanol. As an illustration, *H. uvarum* can only generate 1.0 vol% of ethanol from 19.0 g/L of sugar, and its involvement in inoculation with *S. cerevisiae* has been noted to decrease the ethanol concentration in wine by 1.21 vol% [49,50]. The current work revealed *H. uvarum* and *P. fermentans* have the potential to reduce the concentration of ethanol through mixed fermentation by approximately 0.26–0.67 vol% and 0.36–0.72 vol%, correspondingly (Table 4). The timing of NS inoculation exerted a substantial impact on ethanol output, while different ratios of NS and *S. cerevisiae* barely made a difference. The statistics showed sequentially rather than simultaneously inoculating NS and *S. cerevisiae* decreased ethanol efficiently, in accordance with the findings of You et al. [51]. The reduction in ethanol yield may be due to the con-

sumption of glucose by NS yeasts through oxidative metabolism, leading to the conversion of glucose to compounds, such as glycerol instead of ethanol [52]. Benito discovered that in mixed-culture fermentation involving *P. kluyveri*, the glycerol concentration increased by 0.33–1.30 g/L [53]. We found sequential inoculation (WSP1 and WSP10) of *P. fermentans* and *S. cerevisiae* elevated the glycerol content by 0.43–0.47 g/L.

We also discovered there were alterations in the levels of several aromatic compounds in wine. Inoculation of grape juice with *H. uvarum* cells at a high proportion (WSH10 and WCH10) with *S. cerevisiae* increased isoamyl acetate by 10.9–14.6% and 2-phenethyl acetate by 15.1–30.1%, thereby enhancing the floral and fruity aromas of the wine (Table 5). Analogous outcomes have been described in earlier investigations [53]. However, the participation of *H. uvarum* in mixed fermentations raised the concentration of ethyl acetate, triggering oxidative odors in Italian Riesling wines. Capece et al. [54] explained that the escalation in acetate esters content could be related to some extracellular enzymes secreted by this species. On the other hand, Domizio et al. [6], Anfang et al. [14], and Clemente-Jimenez et al. [55] registered that *Pichia* spp., especially *P. fermentans*, could augment the levels of higher alcohols and ethyl esters in white wine, giving it a richer floral and tropical fruit flavor. We noticed that excessive *P. fermentans* in sequential inoculation significantly improved the contents of ethyl caproate, ethyl caproate, ethyl nonanoate, isobutanol, 2,3-butanediol, and phenethyl alcohol (Table 5).

## 5. Conclusions

In this study, all mixed fermentation schemes can reduce the alcohol content and increase the wine glycerol content, which improves the taste of wine. Regarding the aroma, *H. uvarum* can increase the content of acetate substances in wine and increase the floral of the wine, but a high proportion of inoculation will bring undesirable smells described as oxidation and yeast. *P. fermentans* can increase the contents of esters and higher alcohols in wine, which improve the aroma of fruit and cream. Ultimately, we found mixed fermentations of WSP10 and WCP10 have a better effect, which produces more volatile aroma substances and glycerol content, and increases the total amount of esters by 49.4% and 56.5% compared with the CK, respectively., Additionally, the sensory evaluation score is significantly higher than the CK. Therefore, it is considered that these two mixed fermentation and inoculation methods are more suitable for the vinification of 'Italian Riesling' dry white wine at the eastern foothill of Helan Mountain. Further work is needed to refine some of the fermentation conditions, including the sulfur dioxide, fermentation temperature, and yeast assimilable nitrogen content to provide some reference for improving the serious homogenization of domestic wine and to provide a theoretical basis for the application of mixed fermentation in industry production.

**Author Contributions:** Conceptualization, J.Z. and H.X.; methodology, H.X.; software, Z.Z., H.X. and L.S.; validation, H.X. and Z.Z.; formal analysis, H.X.; investigation, Z.Z. and H.X.; resources, H.X. and L.S.; data curation, H.X. and Z.Z.; writing—original draft preparation, H.X.; writing—review and editing, H.X., Z.Z. and Q.Z.; visualization, Z.Z., L.S. and Q.Z.; supervision, H.X. and J.Z.; project administration, J.Z. and Z.Z.; funding acquisition, J.Z. All authors have read and agreed to the published version of the manuscript.

**Funding:** This research was funded by [Ningxia Hui Autonomous Region Key Research and Development Program] grant number [2020BCF01003].

**Data Availability Statement:** Data are contained within the article.

**Conflicts of Interest:** The authors declare no conflict of interest.

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
