# Peer review of "Effects of Mixed Fermentation on the Aroma Compounds of ‘Italian Riesling’ Dry White Wine in Eastern Foothill of Helan Mountain"

_fermentation, doi:10.3390/fermentation9030303_

Round 1
Reviewer 1 Report
The authors studied the effect of mixed fermentation of non-Saccharomyces strains and Saccharomyces cerevisiae on the aroma quality of ‘ Italian Riesling ’ wine in the Eastern Foothill of Helan Mountain, and determined the most optimum process of mixed fermentation. The research content of this paper is very meaningful. It is acceptable for publication after some modifications.
1 Abstract: The abstract is ok. Keywords: the keywords is repeated with the title, please revise it.
2 Introduction: This part needs to be re-written. Some arguments are poorly supported by literature, The first paragraph is too long and confusing.
3 Materials and Methods: Please provide the specific composition and proportion of these media; “Physicochemical indexes such as alcohol content, volatile acid, reduced sugar and titratable acid were determined according to GB/T 15038-2006”, the authors should give a detailed description of these detection methods.
4 Results and discussion: All the results are described in detail, but the discussion part is insufficient.
5 Conclusion: The conclusion needs to be revised and condensed.
6 The reference section requires careful revision of format and content, for example, some references do not have page numbers.
7 There are many grammatical errors in this article. Please polish the language of this article.
Reviewer 2 Report
The paper “Effects of Mixed Fermentation on the Aroma Compounds of ‘ Italian Riesling ’ Dry White Wine in Eastern Foothill of Helan Mountain” deals with the use of non-Sacch yeast strains in mixed and sequential fermentation with Sc in a specific grape must. It is a topic widely covered in the last decade and obviously it is of interest. The work is presented in a somewhat confused way both from the point of view of the experimental plan and for the English language which often makes the concepts unclear and understandable. The authors conclude the manuscript saying that “these two mixed fermentation and inoculation methods are more suitable for the brewing of Italian Riesling dry white wine at the eastern foothill of Helan Mountain”. There is confusion with brewing (process of beer production), while the work deals with the fermentation of grape must (winemaking = process of wine production). Apart from this, it should be emphasized that the validation of a starter culture for industrial use needs the achievement of results obtained on pilot or winery scale, i.e. in high volumes. It is reported that fermentations carried out on a laboratory scale give results that are not really comparable with the real scale in the winery. Thus, laboratory scale data are useful and important to provide the comparative differences between the behavior of different starter cultures, which in this case are mixed or sequential cultures of non-Saccharomyces-Saccharomyces cerevisiae strains.
Going into detail on some points:
- In the methods section Fermentation strategies can be presented in a table with the different points in order to offer more clarity. Furthermore, the volume in which the fermentations were carried out must be specified, it is not clear from the text. Also more details are necessary on the inoculum conditions.
- In the Results: Point 3.1. Morphological characteristics of the yeast colonies
“The Wolerstein (Correct in Wallerstein) Laboratory nutrient agar medium (Wallerstein Laboratory Nutrient Agar), also known as WL medium, was initially used to monitor yeast populations during fermentation. Many studies[21] shown that most yeast species commonly found during wine fermentation can be distinguished by colony color or morphology on WL medium. The morphological characteristics of different yeast on WL solid medium varied[22]. During the mixed fermentation of wine, different yeast were inoculated, and WL medium can be used to quickly identify most of the relevant yeast during fermentation by the characteristic morphology of each yeast colony, and classify each yeast.” This part needs to be improved and inserted in the Methods, with the references and also the table 2 with the behavior of the strains.
- Table 2 Correct: Species, not genera
- Figure1 Add the Standard Deviation in the point of the curves and add details in the legend, which is not so clear
- Figura 2 Add the Standard Deviation in the point of the curves and add details in the legend, which is not so clear
- Table 4 Aroma compounds: in addition to the aroma descriptors add the threshold value of the compounds
- Figure 3 is meaningless, it needs to be eliminated
Reviewer 3 Report
The manuscript needs major revisions.

Round 2
Reviewer 2 Report
The work has been improved
Reviewer 3 Report
The manuscript has been revised properly.